# Epidemiology and Genetic Analysis of SARS-CoV-2 in Myanmar during the Community Outbreaks in 2020

**DOI:** 10.3390/v14020259

**Published:** 2022-01-27

**Authors:** Wint Wint Phyu, Reiko Saito, Keita Wagatsuma, Takashi Abe, Htay Htay Tin, Eh Htoo Pe, Su Mon Kyaw Win, Nay Chi Win, Lasham Di Ja, Sekizuka Tsuyoshi, Kuroda Makoto, Yadanar Kyaw, Irina Chon, Shinji Watanabe, Hideki Hasegawa, Hisami Watanabe

**Affiliations:** 1Division of International Health (Public Health), Graduate School of Medical and Dental Sciences, Niigata University, Niigata 951-8510, Japan; jasmine@med.niigata-u.ac.jp (R.S.); waga@med.niigata-u.ac.jp (K.W.); irinachon@med.niigata-u.ac.jp (I.C.); 2Infectious Diseases Research Center of Niigata University in Myanmar (IDRC), Graduate School of Medical and Dental Sciences, Niigata University, Niigata 951-8510, Japan; sumonkyawwin@gmail.com (S.M.K.W.); dr.naychiwin@gmail.com (N.C.W.); lashamdidi2010@gmail.com (L.D.J.); hwatanabe@med.niigata-u.ac.jp (H.W.); 3Division of Bioinformatics, Graduate School of Science and Technology, Niigata University, Niigata 950-2181, Japan; takaabe@ie.niigata-u.ac.jp; 4National Health Laboratory, Department of Medical Services, Dagon Township, Yangon 111-91, Myanmar; drhtayhtaytin@gmail.com (H.H.T.); ehhtoodr@gmail.com (E.H.P.); 5Pathogen Genomics Center, National Institute of Infectious Diseases, Tokyo 189-0002, Japan; sekizuka@nih.go.jp (S.T.); makokuro@niid.go.jp (K.M.); 6Respiratory Medicine Department, Thingangyun Sanpya General Hospital, Yangon 110-71, Myanmar; ydkkyaw@gmail.com; 7Influenza and Other Respiratory Virus Research Center, National Institute of Infectious Diseases, Tokyo 189-0002, Japan; sw@nih.go.jp (S.W.); hasegawa@nih.go.jp (H.H.)

**Keywords:** SARS-CoV-2, COVID-19, community transmission, case fatality rate, reproductive number, molecular epidemiology, whole-genome sequences

## Abstract

We aimed to analyze the situation of the first two epidemic waves in Myanmar using the publicly available daily situation of COVID-19 and whole-genome sequencing data of SARS-CoV-2. From March 23 to December 31, 2020, there were 33,917 confirmed cases and 741 deaths in Myanmar (case fatality rate of 2.18%). The first wave in Myanmar from March to July was linked to overseas travel, and then a second wave started from Rakhine State, a western border state, leading to the second wave spreading countrywide in Myanmar from August to December 2020. The estimated effective reproductive number (*R_t_*) nationwide reached 6–8 at the beginning of each wave and gradually decreased as the epidemic spread to the community. The whole-genome analysis of 10 Myanmar SARS-CoV-2 strains together with 31 previously registered strains showed that the first wave was caused by GISAID clade O or PANGOLIN lineage B.6 and the second wave was changed to clade GH or lineage B.1.36.16 with a close genetic relationship with other South Asian strains. Constant monitoring of epidemiological situations combined with SARS-CoV-2 genome analysis is important for adjusting public health measures to mitigate the community transmissions of COVID-19.

## 1. Introduction

The coronavirus disease2019 (COVID-19) pandemic caused by severe acute respiratory syndrome coronavirus 2 (SARS-CoV-2) continues to be a major global concern [1]. In late 2019, the first cluster of people with severe respiratory infections and pneumonia was reported in the Hubei Province of China [2,3,4]. These are highly infectious viruses possessing an unprecedented speed of transmission. Globally, during the first six months of 2020, the number of confirmed cases reached more than seven million, of which there were more than 400,000 attributable deaths [5,6].

The virus causing COVID-19 induces common symptoms such as fever, cough, shortness of breath, and fatigue [7,8]. Initial estimates of the case fatality rates (CFR) ranged from 3.4% to 6.6%, which is lower than that of SARS and Middle East Respiratory Syndrome (9.6% and 34.3% respectively) [3,9]. As of 4 February 2020, the epidemiological estimate of the basic reproductive number from the beginning of the epidemic (i.e., 30 January 2020) to 1 February 2020, was 2.2 (range, 3.6–5.8), and the estimated doubling time of the epidemic was a mean of 3.6 days (varying from 1.0 to 7.7) in Wuhan, China [10]. According to the World Health Organization [1], COVID-19 reached the pandemic phase on 11 March 2020, at which point it achieved a very high spreading rate with a reproductive number higher than that of influenza because of a lack of preceding immunity [11]. Correspondingly, governments worldwide implemented a range of measures aimed at limiting people’s movements and interactions to attempt to slow down the transmission of COVID-19 and dampen the severity of the epidemic trajectory. Data from the epidemic in Wuhan, China suggest that the time-varying effective reproductive number (*R**_t_*) declined in February 2020 after authorities-imposed lockdowns [12].

In Myanmar, the first two confirmed cases of COVID-19 were reported on 23 March 2020, in people who had recently returned from the United States of America (USA) and the United Kingdom (UK). Subsequently, Myanmar experienced two community outbreaks in 2020. As of 31 December 2020, there were 33,917 confirmed cases and 741 deaths in Myanmar [13].

This is the first comprehensive report to describe the epidemiological situation in Myanmar in 2020. We have used the official surveillance data of COVID-19 in Myanmar, described characteristics of the two epidemic waves with case fatality rates, and estimated the effective reproductive number for the epidemics. In this study, the SARS-CoV-2 whole-genome sequences from 10 patients in Myanmar collected in August 2020 at the beginning of the second wave were investigated to identify the genetic diversity among the viruses together with 31 sequences [14] deposited in the public genetic database from Myanmar during 2020, to understand the molecular epidemiology of the SARS-CoV-2 virus in Myanmar.

## 2. Materials and Methods

### 2.1. Data Collection

We retrieved publicly available data from the website of the Ministry of Health and Sports (MoHS), Coronavirus Disease 2019 (COVID-19) Surveillance Dashboard, Myanmar, of laboratory-confirmed cases and death cases between 23 March 2020, and 31 December 2020, which covered the daily situation of COVID-19 [13]. Then, we constructed epidemic curves using time-series surveillance data from the MoHS website to clarify the epidemic situation of COVID-19 in Myanmar. 

### 2.2. Case Definition

The MoHS, Myanmar followed WHO guidelines and established its clinical management guidelines for the detection and treatment of COVID-19 cases, in which patients with a confirmed diagnosis of COVID-19 were defined as having symptoms of respiratory infection (fever, cough, or dyspnea), non-specific symptoms (fatigue, headache, myalgia, loss of smell, loss of taste, sore throat, coryza, nausea, vomiting, diarrhea, and altered mental status), or a positive test for SARS-CoV-2 via a real-time PCR (RT-PCR) test or an rapid antigen diagnostic test (e.g., STANDARD Q COVID-19 Ag, SD Biosensor, Seoul, Korea) based on respiratory tract samples [15]. Imported cases were those suspected to have acquired the infection outside of Myanmar [9]. Contact history refers to when a person has household contacts or contacts made in crowded or closed settings (e.g., long-term living facilities, prisons, shelters, hostels, gyms, and meeting rooms) wherethe confirmed or suspected COVID-19 case(s) remained, especially in settings with poor ventilation. COVID-19 death was defined as a death resulting from a clinically compatible illness in a probable or confirmed COVID-19 case unless there is a clear alternative cause of death not related to the COVID-19 disease (e.g., trauma) [15]. 

### 2.3. Time-Varying Effective Reproductive Number

To investigate the COVID-19 transmission intensity throughout an epidemic, the *R_t_* was estimated from the incidence time-series surveillance data by applying the same methodology as Cori et al. [16], in which the *R_t_* can be estimated by the ratio of the number of new infections generated at time step *t* (*I_t_*) to the total infectiousness of the infected individuals at time *t*, given by $\sum_{s=1}^{t}I_{t-a}w_{s}$, where the sum of infection incidence up to the time step *t*−1 is weighted by the infectivity function *w_s_*. *R_t_* is the average number of secondary cases that each infected individual would infect if the conditions remained as they were at time *t* and is thus commonly used to characterize pathogen transmissibility during an epidemic. In general, a *R_t_* value greater than one indicates that more cases are occurring and the infection is spreading, while a *R_t_* of less than one indicates that the spread of the infection is decreasing. Theoretically, we need information regarding the generation time, which is defined as the period between the infection of the index and the next case, but this information is usually difficult to ascertain. Instead, the *R_t_* value can be adjusted to include the serial interval (SI), which is defined as the interval between the onset of the index and the next case, as an infectivity function, assuming a gamma distribution [16,17]. We used the SI (mean SI: 4.7 days, Standard Deviation SI: 2.9 days) described by Nishiura et al. [18], and our time-varying estimates were made with a seven-day sliding window. Statistical estimation was performed using R statistical programming software ver.3.6.3 (https://www.r-project.org/ (accessed on 12 June 2021)) with the “EpiEstim” package [16,17].

### 2.4. Laboratory Confirmation of COVID-19 

Nasopharyngeal and oropharyngeal swabs from suspected patients were collected at quarantine centers or hospitals. Each swab was placed in a viral transport medium and sent to the National Health Laboratory (NHL), Yangon, Myanmar for laboratory confirmation of COVID-19 infection.

#### 2.4.1. RNA Extraction and RT-PCR 

SARS-CoV-2 viral RNA was extracted from 140 μL of nasopharyngeal and oropharyngeal swabs suspended in the viral transport media using a QIAamp Viral RNA mini kit (QIAGEN, Hilden, Germany) following the manufacturer’s instructions. RT-PCR using a Novel Coronavirus (2019-nCoV) Nucleic Acid Diagnostic Kit (PCR-Fluorescence Probing) (Sansure Biotech Inc., Changsha, China) was performed according to the kit protocol to detect the ORF1a/b and N genes of SARS-CoV-2, and amplification was performed using the Bio-Rad CFX96 Real-Time System (Life Science Research, Franklin, NJ, USA). Another test to detect SARS-CoV-2 was conducted at the NHL using the commercially available cobas^®^ SARS-CoV-2 reagents (Roche Diagnostics, Indianapolis, IN, USA) with the Cobas 6800 System, which is a fully automated RT-PCR testing system for sample preparation, nucleic acid extraction and purification, PCR amplification, and detection. In September 2020,10 RT-PCR positive samples were sent from the NHL to the National Institute of Infectious Diseases (NIID), Japan for whole-genome sequencing of the SARS-CoV-2 viruses.

#### 2.4.2. Whole-Genome Sequencing of SARS-CoV-2 

Whole-genome sequencing was performed at the NIID, Japan. The whole-genome sequences of SARS-CoV-2 were obtained using the PrimalSeq protocol to enrich the cDNA of the SARS-CoV-2 genome using multiplex RT-PCR amplicons with a multiplexed PCR primer set, as proposed by the Welcome Trust ARTIC Network. We used the primer for multiplex PCR amplification selected by Itokawa et al. [19]. The PCR products from the same clinical sample were pooled, purified, and subjected to Illumina library construction using the QIAseq FX DNA Library Kit (QIAGEN, Hilden, Germany). The Illumina NextSeq 2000 platform (Illumina, San Diego, CA, USA) was used to sequence the indexed libraries. Detailed methods for next-generation sequencing are reported in the literature [19].

#### 2.4.3. Phylogenetic Analysis

To clarify the genotypes of strains in Myanmar, 10 sequences in this study and an additional 31 sequences already registered in the public database were analyzed by Nextstrain to find out Nexstrain and GISAID clades (https://clades.nextstrain.org (accessed on 30 November 2021)) [14,20,21]. The 10 whole genomes sequenced in the current study have been deposited in the GISAID EpiCoV coronavirus SARS-CoV2 database (www.gisaid.org (accessed on 12 June 2021) (Appendix A), and 31 strains previously submitted from the Department of Medical Research (DMR), Myanmar, were downloaded from the GISAID (EPI_ISL_896572- EPI_ISL_896601) (Appendix A) [22]. The Phylogenetic Assignment of the Named Global Outbreak Lineages (PANGOLIN) web application was also used to determine PANGOLIN lineages for the 41 strains (https://pangolin.cog-uk.io/ (accessed on 12 June 2021)) [23].

To further characterize the genomic shifts over time and trace the transmission pattern of how SARS-CoV-2 viruses were circulating in Myanmar, phylogenetic analysis using Nextstrain version 8.0 with default setting was performed against 6195 global strains which were sampled collected from December 2019 to September 2020, and 41 Myanmar strains including 10 strains obtained in this study (https://github.com/nextstrain/ncov (accessed on 30 November 2021)) [20,21]. Multiple alignments for the full-length genome sequences of SARS-CoV-2 were performed using MAFFT v.7.475 [24]. The removal of spurious sequences or poorly aligned regions from multiple sequence alignments was performed using trim AL v.1.4. 

In addition, we performed phylogenetic analysis, using a total of 1320 genomic sequences, including 10 strains obtained in this study, additional 31 GISAID registered sequences from the other group in Myanmar, 509 global strains, and 770 Asian strains from December 2019 to September 2020. The Asian strains were constructed from neighboring countries of Myanmar, and the sequence data registered in GISAID by August 2020 were extracted. As a reference sequence, Wuhan/WIV04/2019 (GISAID ID, EPI_ISL_402124), which was used as the official reference sequence by GISAID, was selected. Maximum-likelihood (ML) phylogenetic analysis was performed using IQ-TREE v.2.1.2 with Model Finder and ultrafast bootstrap test parameters [25,26,27], and visualized using iTOL v.6 [28]. All data analysis of constructing whole-genome sequences of SARS-CoV-2 was performed at Niigata University, Japan.

#### 2.4.4. Variant Analysis of SARS-CoV-2 Genome

All distinct base-pair changes and mutations in the viral sequences were evaluated. We also analyzed the cluster variation based on amino acid changes. We downloaded the ancestral Wuhan reference strain Wuhan/WIV04/2019 (EPI_ISL_402124) to identify genetic variants in the selected Myanmar strains. The sequences of the 5ʹUTR gene (nucleotides 1–265), ORF1ab protein (nucleotides 266–21,555), including NSP2 (nucleotides 806–2719), NSP6 (nucleotides 10,973–11,842), NSP8 (nucleotides 12,092–12,685), RNA dependent RNA polymerase (RdRP/NSP12) (nucleotides 13,442–16,236), and NSP14 gene (nucleotides 18,040–19,620), surface spike (S) glycoprotein (nucleotides 21,563–25,384), ORF3a protein/NS3 (nucleotides 25,393–26,220), Envelope (E) protein (nucleotides 26,245–26,472), Membrane glycoprotein (M) (nucleotides 26,523–27,191), Nucleocapsid phosphoprotein (N) (nucleotides 28,274–29,533), and ORF8 to ORF10 (nucleotides 27,894–29,657) were selected to analyze nucleotide and amino acid variation.

### 2.5. Ethical Considerations

All study participants were enrolled, and samples were collected in NHL, Myanmar following the MoHS government-approved protocol, laboratory sample collection guideline version 2 (https://mohs.gov.mm/ (accessed on 12 June 2021)) [13]. Oral informed consent was obtained from all participants for SARS-CoV-2 testing following the local infection control policy during outbreaks in Myanmar. Because of the study design, as this study was conducted under the national surveillance in the whole country under the control measurement policies of the Myanmar government against COVID-19, and the ongoing public health emergency to control the outbreak, as well as the importance of sharing the research findings and bridging the knowledge gaps, ethical approval was waived by an institutional review board. All methods were performed under the relevant guidelines and regulations, as all samples were anonymous. The datasets used in our study were concealed and fully anonymized before experimentation. The datasets generated during this study are available upon request from the corresponding author. 

## 3. Results

### 3.1. Epidemic Situation of SARS-CoV-2 in Myanmar 

Since the WHO declared COVID-19 a public health emergency of international concern on 30 January 2020, Myanmar began preparing for a possible outbreak by forming the National Level Central Committee for COVID-19 Prevention, Control, and Treatment. In Myanmar, the first confirmed case was reported in Chin State (western Myanmar) on 23 March 2020, in a patient who had recently returned from the USA. Of the 20 confirmed cases between 23 March and 3 April 2020, 12 were imported from other countries, whereas eight cases were infected by local transmission. During the second week of April, the first cluster, a religious cluster linked with 70 cases, occurred in the Yangon Region (Figure 1), which is the largest city in Myanmar (Appendix A). Therefore, the Yangon Region government announced lockdown and stay-at-home measures on 18 April 2020. In April, a total of 136 cases were reported as laboratory-confirmed cases. The number of confirmed COVID-19 cases dramatically decreased from the end of April through July as a result of the rapid containment procedure of stay-at-home measures and effective responses (Figure 2A). During the first wave, the number of deaths (six) and CFR (1.7%; 6/353) was very low from March to July (Figure 2B).

### 3.2. Second Wave of SARS-CoV-2 Outbreak in Myanmar

In Rakhine State, Myanmar (Figure 1), a western state bordering with Bangladesh, the first case of COVID-19 was detected on 16 August 2020, approximately five months after the detection of the first imported case. The number of laboratory-confirmed cases increased to 80 between 16 August and 21 August 2020. The MoHS officially declared a stay-at-home order in Rakhine State on 27 August 2020, covering the dates of 16 August to 22 September 2020, during which the number of confirmed cases increased to 1175 in Rakhine State (Appendix A). Subsequently, the infection in Rakhine State spread country-wide, resulting in a second wave that was larger than the first wave that occurred in April in Myanmar. Similarly, in Yangon, the number of cumulative confirmed cases increased from 249 in July to 361 in August to 9289 in September and 40,103 in October. The nationwide data showed that the case number peaked in the middle of October but decreased thereafter, probably due to public health measures such as travel restriction, a stay-at-home order, and a mask and face shield campaign. There was a re-surge in the middle of November but gradually decreased toward the end of December. As of 25 December 2020, the number of confirmed cases was 81,214 in Yangon and 4102 in Rakhine (Appendix A). From August to December, 2682 deaths occurred in Myanmar (Figure 2). Compared with the CFR of the first wave (1.7%; 6/353), that in the second wave was 2.2% (741/33,917). During the second wave, the CFR was almost steady throughout September (2.4%; 304/12,486), October (2.4%; 927/39,333), November (1.9%; 704/38,007), and December (2.2%; 741/33,917) (Figure 2). 

### 3.3. Effective Reproductive Number of SARS-CoV-2 

To assess the effectiveness of COVID-19 control measures and to understand the infection status in Myanmar, we estimated the time-varying *R_t_* (Figure 2C). Therefore, it was suggested that the estimated median initial *R_t_* of the first wave observed around 15 April 2020 may have reached 5.99 (95% credible interval (CI): 4.84–7.31). Subsequently, when the epidemic expanded rapidly in the second wave around 25 August 2020, the estimated median *R_t_* reached 7.54 (95% CI: 6.69–8.47), after which the *R_t_* gradually decreased because of rapid containment procedures and effective control measures (e.g., lockdown). The *R_t_* continued to decrease from September onwards, remaining around one at the end of December.

### 3.4. Demographic Characteristics of 41 COVID-19 Patients 

During the second wave in Myanmar, in total, 10 samples of confirmed COVID-19 patients detected at NHL in Myanmar at the beginning of the second wave between 21 August and 25 August 2020 were transported to the NIID, Japan in the first week of September 2020 for whole-genome sequencing. These samples were selected because they were collected during the early phase of the second wave and showed higher RNA concentrations, attaining C_t_ values of<30 by RT-PCR at the time of testing in the NHL, Myanmar.

Of the 10 samples, nine patients were from the quarantine center of Rakhine State, and one patient was from Yangon Region (Table 1). Most of the samples were from Rakhine State because the local transmission was further progressed in Rakhine State than Yangon Region at the time of sample shipment. The extracted RNA from the clinical samples was shipped using an international courier. The patients were adults, with ages ranging from 20 to 71 years. Only four patients had a history of contact with known COVID-19 cases. Five patients had a history of domestic travel before symptom onset. Only one patient developed mild symptoms of fever and loss of smell, whereas the rest of the patients remained asymptomatic. All 10 patients recovered without complications.

According to the previously published report from other group in Myanmar, they analyzed the demographic characteristics of 31 COVID-19 patients from April to September 2020. Among them, 21 patients were from Yangon City, and ten patients were from the Rakhine States [14]. The patients were adults, ages ranging from 21 to 84 years, and female (54.8%, 17/31) were similar number of male cases (45.2%, 14/31). Out of 31 cases, 14 local transmission cases had a history of contact with known COVID-19 patients, but no traveling history to foreign countries. The remaining 17 patients had no contact history with a known positive case [14]. Additionally, the clinical symptoms were extracted from the matching cases from the official website of Ministry of Health [13]. As the detailed information on the Ministry of Health website is limited, we could obtain the clinical symptoms of only 5 out of 31 patients (Appendix A). Out of five cases, four were asymptomatic, and only one patient had sneezing and loss of taste, in which nine of them were deceased cases in Yangon, especially in the COVID-19 patients in September 2020.

### 3.5. Genome Analysis of SARS-CoV-2 Viruses in Myanmar

The genetic analysis of 41 strains (10 from this study and 31 downloaded from GISAID) showed that first, five Myanmar sequences during April to early May 2020 belonged to GISAID clade O, Nextstrain clade 19A, and PANGOLIN B.6 (Figure 3A and Appendix A) [14]. Thereafter, the genotypes diversified and changed into GISAID clade G (Nextstrain 20A; PANGOLIN B.1.210), GR (Nextstrain 20B; PANGOLIN B.1.1 and B.1.1.174), and GH (Nextstrain 20A; PANGOLIN B.1.36) during May and June 2020, mostly related to the overseas travels such as India and China (Appendix A) [14]. The genome analysis during the second wave in August–September showed that a total of 29 sequences, 10 sequences in this study and 19 sequences from the other study [14], belonged to GISAID clade GH, Nextstrain clade 20A, and the PANGOLIN lineage B1.36.16 (Figure 3A, Appendix A). The time-aware phylogeny and phylogeographic analysis by Nextstrain using 41 Myanmar strains and 6195 global strains showed that there are multiple introductions from other foreign countries, such as China at the beginning of the first wave and then changed to India and Bangladesh in the second wave (Figure 3B–D, Appendix A). Moreover, strains detected in Myanmar were the same genetic cluster as the virus strains from India and Bangladesh registered from May to October 2020, showing the strong relationship of the strains with the two countries (Figure 3C, Appendix A).

We additionally performed a phylogenetic tree analysis with 41 sequences from Myanmar and 1279 global strains. At the beginning of the first wave in Myanmar, clade O strains in the previous study in Myanmar were very close to the reference sequence from Wuhan, China (Figure 4A). Additionally, the three clade GR strains from Myanmar in May and August were similar to the virus strains from USA and Egypt registered from April to July (Figure 4B). Moreover, one of the GH clades sequences in the first wave in Myanmar in May 2020 was close to the virus strains from India registered in May 2020 (Figure 4C). During the second wave, all 29 strains in clade GH in Myanmar were most similar to those in clade GH from India and Bangladesh during July and August 2020 (Figure 4D). 

### 3.6. Identification of Clades and Genetic Diversity in SARS-CoV-2 Genomes 

We performed a comprehensive variant analysis of the 10SARS-CoV-2 strains in this study from Myanmar. In total, we identified mutations in 29 nucleotide sites and five major amino acid mutations in all the selected isolates, as compared with the reference strain Wuhan/WIV04/2019. 

Single nucleotide changes in the 5ʹUTR (C241T), ORF1b (C14408T and C3037T), S (A23403G), and ORF3a (A25563/Q57H) were identified in the 10 Myanmar-sequenced strains (Table 2). The most prevalent mutation in our selected sequenced genomes was a transversion affecting the 23,403-nucleotide adenosine, amino acid substitution from alanine to guanosine (A23403G), defining the G-clade of the SARS-CoV-2 genomes [29]. Specifically, four mutations, C241T, C3037T, C14408T, and A23403G, were observed in all samples (Table 2). Of these, C3037T and C14408T did not affect the amino acid, as they are silent mutations targeting the 5′UTR in position 241. Mercatelli and Giorgi [29] reported that these four mutations almost always co-occur in the major clade G. In addition, the genome mutations D614G in Spike and Q57H in ORF3a were mostly found in the GH clade. Therefore, the Myanmar sequences can be clearly defined by the GH clade.

Of the five amino acid mutations, three major mutations, L37F(G11083T) in the putative transmembrane domain (NSP6) region of the ORF1ab protein, D614G(A23403G) in the S, Q57H(G25563T) in the ORF3a protein, and S194L(C28854T) in the N protein, were observed in all the selected Myanmar strains (100%,10/10). Additionally, S369F (C19145T) in the NSP14 region of the ORF1ab protein was found in three strains (33.3%, 3/10). The rest of the genome was conserved, with no significant amino acid substitutions. 

No amino acid substitutions were detected in the E or M proteins, indicating that these proteins are highly conserved among these viruses.

## 4. Discussion

In Myanmar, the first confirmed COVID-19 case was reported on 23 March 2020, and the first death case was reported on 31 March. The number of positive COVID-19 cases increased rapidly in April, forming the so-called first wave, after which the case occurrence dramatically decreased at the end of April until July in 2020. However, a community outbreak occurred in the middle of August in Rakhine State, then outbreaks spread nationwide, forming a second wave. The official surveillance data showed that the first wave was more related to international travels and local gatherings than the second wave. Case fatality rates of the two waves did not differ, at around 2%, despite the fact the second wave was bigger than the first. The estimated effective reproductive number (*R_t_*) reached 6–8 at the beginning of each wave and gradually decreased as COVID-19 infections spread to the community. We performed a comprehensive whole-genome analysis of 10 Myanmar SARS-CoV-2 strains together with previously registered 31 strains and identified the first and the second waves were caused by the different genotypes. Specifically, the second wave was caused by PANGOLIN lineage B1.36.16, GISAID clade GH possessing D614G in Spike protein. The phylogeographic analysis suggested that the Myanmar strains in the second wave had a close relationship with other south Asian strains, such as India and Bangladesh.

In Myanmar, the first confirmed case and first death were identified as having been imported from other countries, such as the USA and the UK from March to April 2020. The first wave in Myanmar was associated with a religious cluster in the Yangon Region during April 2020. A similar cluster occurred in a neighboring county in Thailand. In March 2020, an increased number of imported patients were reported in Thailand, and subsequently, a group that had undertaken a religious pilgrimage from the southern part of Thailand formed a cluster [30]. South Korea also announced thousands of COVID-19 cases within a few days in late February 2020. This surge in cases centered mostly around one main cluster from the Shincheonji church in Daegu city, triggering a drastic escalation of the South Korean spread of confirmed cases of theSARS-CoV-2infection [31]. These outbreaks indicate that indoor gatherings, such as religious events, which are typically held with people in very close proximity and include physical contact and chanting, easily cause clusters as a result of the droplet, aerosol, and contact infections [32]. Upon local surges in the first wave, the Myanmar government instituted the mandatory closure and restriction of public places to control local transmission. After land border closure and the suspension of all international flights, the number of cases decreased from May to July 2020. 

The second wave that occurred in mid-August 2020 started in Rakhine State, which is the west coast region of Myanmar. The daily laboratory-confirmed cases were the highest in Sittwe, which is the capital of Rakhine State, and were identified as community-transmitted cases. According to the estimation of median *R_t_,* being 7.54, the epidemic increased rapidly in the second wave around 25 August 2020. Subsequently, as the number of newly confirmed cases increased in Rakhine State, the MoHS officially announced a stay-at-home order in Rakhine State on 20 August 2020, and domestic travel was banned from August to the middle of December, according to the COVID-19 situation reports [13]. Overall, local transmission became much higher than imported cases in the second wave, according to the contact tracing implemented by the MoHS. Around the same time, the number of daily new COVID-19 confirmed cases in the neighboring countries of India and Bangladesh increased significantly. Specifically, on 23 August 2020, Bangladesh became the country with the third-highest incidence in the world [33]. Following the local outbreaks in a western border state, the case number surged in a major city, Yangon, in September, and the government implemented various public health measures including lockdown, stay-at-home order, mask-wearing campaigns, and ban of mass gatherings. The number of COVID-19 cases was high during October and November, but gradually decreased, as shown by a decrease of *R_t_* by the end of December 2020.

Whole-genome analysis showed that the genotypes of the viruses that caused the first and second waves in Myanmar were different. The previous report from the other group in Myanmar identified that nearly all of the sequences from the local cases in the early pandemic spread in April 2020 were PANGOLIN B.6 lineage (GISAID clade O and Nextstrain 19A) [14]. This result indicates that the first wave was caused by the similar lineage observed in the first wave in Malaysia, wherein the Tablighi Jamaat religious mass gathering held in Kuala Lumpur between 27 February and 3 March 2020, had lineage B.6. After the community circulation in April, various genotypes were detected in Myanmar during May and June from overseas returnees, presumably reflecting the viruses circulated in each area.

During the second wave from August to September 2020, Myanmar sequences belonged to the clade GH (Nextstrain clade 20A or PANGOLIN lineage B1.36.16). The first detection of GH (mutations D614G in the Spike and Q57H in ORF3a) subclade was observed in the USA at the end of February 2020 [29]. The most prevalent genome in the USA from March to December 2020 was GH, as it was in Israel and Saudi Arabia [29]. The G- and G-derived clades (GH and GR) reached Asia in March 2020, becoming the dominant viral population worldwide until December 2020 [34]. Our Nextstrain analysis and ML phylogenetic analysis showed that 29 SARS-CoV-2 viruses detected in Myanmar under the GH or PANGOLIN lineage B.1.36.16 clade in August and September were in close relationship with those in Bangladesh and India. Therefore, it was assumed that the SARS-CoV-2 virus that originally circulated in the USA [35] was transmitted through numerous transmission chains into South Asia, and eventually caused the second wave in Myanmar. It was reported that clade GH (20A) started to increase in India in April 2020 [36]. Additionally, clade GH, including PANGOLIN lineage B.1.36.16, circulated in Bangladesh during 2020, but the proportion was less than 10% of total strains analyzed. It is not clear why predominant genotypes differed in neighboring countries although the genetic relationship of viruses was high. We will further monitor the shift of genotypes over time in Myanmar.

The most prominent mutation in our selected sequenced genome was D614G in the S protein. The initial strains in clade O in Myanmar retained D614, similar to the prototype Wuhan strain, but the subsequent strains in clades G, GR and GH changed to the D614G [14]. The D614G mutation causes a change (aspartate to glycine in protein position 614) in the carboxy-terminal region of the S1 domain, which is responsible for the initial entry of the virus into the cell via the ACE2 human receptor [37]. The D614G mutation increases cell entry by acquiring a higher affinity for ACE2 while maintaining neutralization susceptibility [38]. Korber et al. noted that this change was correlated with increased viral loads in COVID-19 patients [39]. This mutation is not in the receptor-binding domain (RBD) of the S protein, but rather exists in the interface between the individual spike protomers that stabilize its mature trimeric form on the virion surface via hydrogen bonding [39]. Therefore, as the D614G mutation does not affect the neutralization by antisera against prototypic [38], it is unlikely to have a major impact on the efficacy of vaccines, some of which exclusively target the RBD [40].

Among the genetic and amino acid variants found in the 10 Myanmar strains in this study and 19 strains reported elsewhere, the nucleotide change C241T found at the silent site (5ʹUTR) may affect the secondary RNA structure [41]. Furthermore, S194L (C28854T) in the N protein was detected in all of 29 Myanmar strains in the second wave. Garvin et al. determined that mutations in this region decrease viral replication [42]. In addition, we identified the Q57H mutation in the open reading frames 3a (ORF3a) in all of the 29 Myanmar strains [14]. These mutations might help the virus to evade the host immune system and may be related to rapid infection kinetics and the spread of SARS-CoV-2 [43]. 

It has been reported that upper respiratory infection, fever, cough, sore throat, and runny nose are the most common symptoms in COVID-19 patients, and some patients develop severe pneumonia [44]. For the 10 patients analyzed herein, the clinical course of COVID-19 was generally mild, as the percentage of asymptomatic patients was 90% and there were no fatalities. Only one patient in this study presented symptoms of fever and loss of smell. According to the previously published paper from the other study in Myanmar, four out of five patients (80%) were asymptomatic, whereas symptoms of 26 patients were not available on the official website. Only one patient had a mild complaint, sneezing, and loss of taste, but there were nine fatal cases in strains registered in GISAID in September 2020 [14]. As of February 2021, loss of smell (anosmia) or loss of taste (ageusia) became the most common symptoms in COVID-19 patients in Myanmar, according to the COVID-19 contact tracing and report from MoHS [13]. A similar finding was reported from Korea, in which acute anosmia or ageusia was observed in 15.3% (488/3191) of patients in the early stage of COVID-19 and their prevalence was significantly more common among females and younger individuals [45]. Anosmia and ageusia seem to be important symptoms, particularly in the early stages of the disease [45]. Therefore, recognizing early signs, such as anosmia or ageusia, might be very helpful for diagnosing COVID-19 and isolating patients [45]. 

In this study, the reported CFR in Myanmar was 1.7% (6/353) in the first wave (as of March 2020), which was almost the same as that of neighboring Thailand (1.9% between January and May 2020), and it remained 2.2% (741/33,917) during the second wave (as of December 2020) [35]. The number of confirmed cases, deaths, and mortality rates (i.e., CFR) related to COVID-19 vary both regionally and by country, and are affected by numerous factors, including health control policies, medical standards, and detection efficiency [30]. India (CFR of 2.8% between January and June 2020) was the most affected south Asian country in 2020 [35], attaining the highest fatality rate, as it is the second-most populous country. Please note that the Bangladesh–Myanmar border is the international border between India and Myanmar. The CFR in Myanmar was lower than that in European countries such as Italy, Spain, France, Netherlands, and the UK (approximately 6.6–11.0%) as of 30 March 2020 [46,47], and western Pacific countries (approximately 2.3–5.5%) as of 9 June 2020 [35]. This might be because the healthcare facilities in Myanmar performed contact tracing, which is a critical strategy in stopping the transmission of COVID-19. 

The major limitations we faced in this study were the limited number of SARS-CoV-2 genomes collected only at the beginning of the second wave, because it was difficult to transport samples internationally from Myanmar to Japan while adhering to the infectious particle restrictions during the COVID-19 pandemic, and we could not install next-generation sequencing in Myanmar because of time and cost limitations. Second, because of the small sample size, a clear picture of clinical symptoms, severity, and viral genetic variation over time, especially a comparison between the first and second waves, was missing. Third, to capture the national status of the epidemic in Myanmar, some patients may be missed because of difficulties in accessing fever clinics and improper sampling in remote areas, causing the under-reporting of data. Fourth, as we could not obtain the illness onset dates of the cases, we calculated the *R_t_* values from the official reported sample collection date. Thus, the estimated *R_t_* may be inaccurate or under-reported.

Nevertheless, this study characterized the epidemic spread pattern of SARS-CoV-2 in Myanmar based on the official nationwide surveillance and whole-genome sequencing data. Combined with the common epidemiological analysis to investigate the infection sources, evaluate the size and severity of community transmissions, the genomic analysis may be extremely useful in identifying strategies and therapies that can help to reduce the burden of the virus, depending on the genotypes circulating [48]. Understanding the epidemic dynamics of SARS-CoV-2 is increasingly important for guiding prevention efforts, and the constant monitoring of mutations is pivotal for tracking the movement of the virus, especially after the emergence and international spread of variants of concern (e.g., alpha, delta, or omicron) that affect infectivity and disease severity, and most importantly, vaccine effectiveness [48]. 

## Figures and Tables

**Figure 1 viruses-14-00259-f001:**
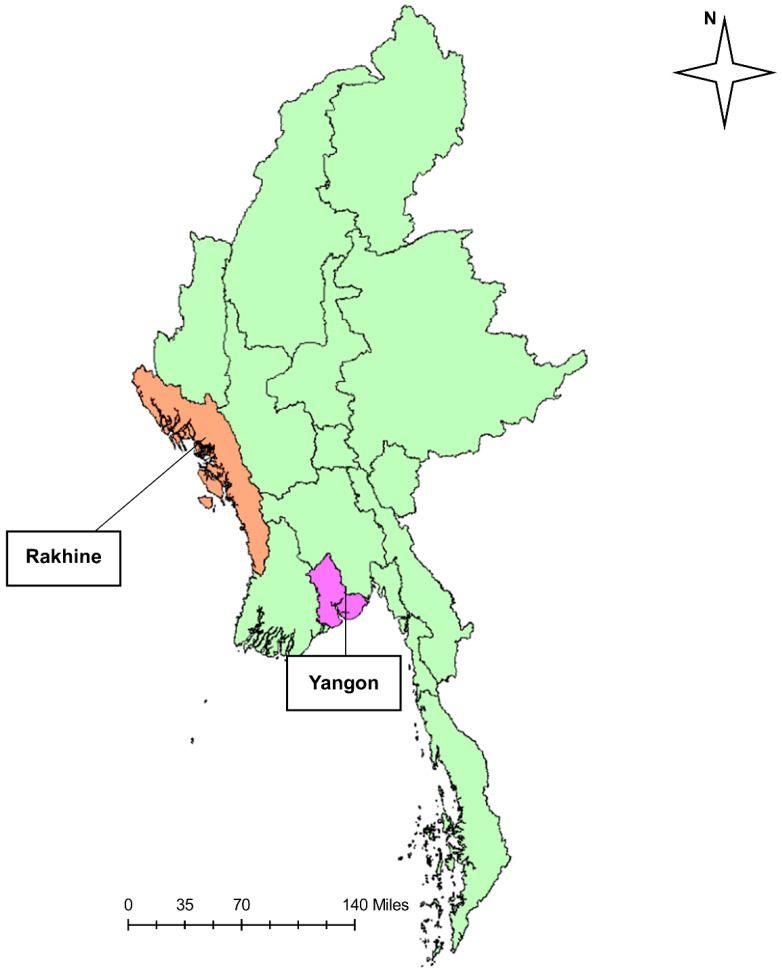
Location of Rakhine State (orange) and Yangon (pink) in Myanmar. Source: This map was generated by ArcMap 10.0 software (ESRI, Tokyo, Japan).

**Figure 2 viruses-14-00259-f002:**
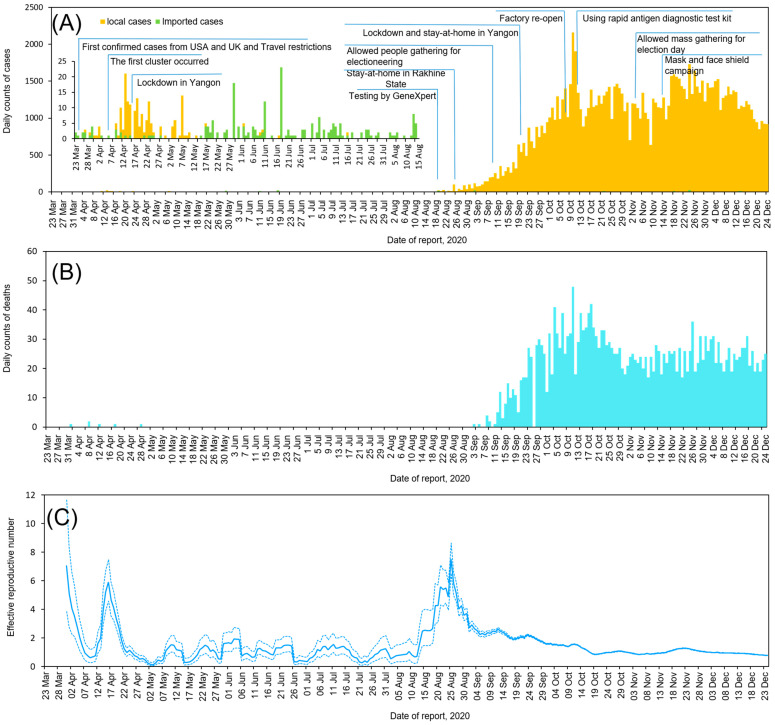
The epidemic situation of coronavirus disease-2019 (COVID-19) in Myanmar from 23 March to 30 December 2020. (**A**) The daily number of laboratory-confirmed cases by reporting date colored as yellow and green and timeline of events. (**B**) The daily number of reported death cases by reporting date colored as light blue. (**C**) Estimated daily reproductive number (*R_t_*). Solid blue indicates time-varying *R_t_* during the study period. Dashed blue lines show 95% credible intervals.

**Figure 3 viruses-14-00259-f003:**
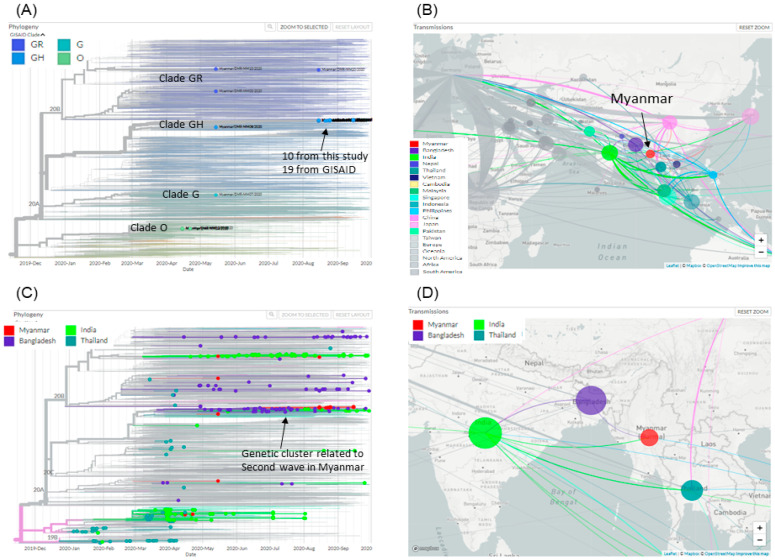
Phylogenetic tree showing the evolutionary relationship of SARS-CoV-2 with 41 Myanmar strains including 10 strains obtained in this study and 6195 global strains collected from December 2019 to September 2020. Sequence data were downloaded from Global Initiative on Sharing All Influenza Data (GISAID), and figures were created by the Nextstrain platform. (**A**) Time-aware phylogenetic tree showing 10 strains from this study (indicated by arrow) and 31 from another study [14], colored by GISAID clade. (**B**) Phylogeographic map showing an estimated geographic relationship among SARS-CoV-2 sequences colored by country. Myanmar was indicated by red. (**C**) Time-aware phylogenetic tree focused on sequences from Myanmar (red), Bangladesh (purple), India (yellow-green), and Thailand (green). The lineage related to local transmission in the second wave in Myanmar in 2020 was indicated by the arrow. (**D**) Phylogeographic map focused on the relationship of the sequences from Myanmar, Bangladesh, India, and Thailand.

**Figure 4 viruses-14-00259-f004:**
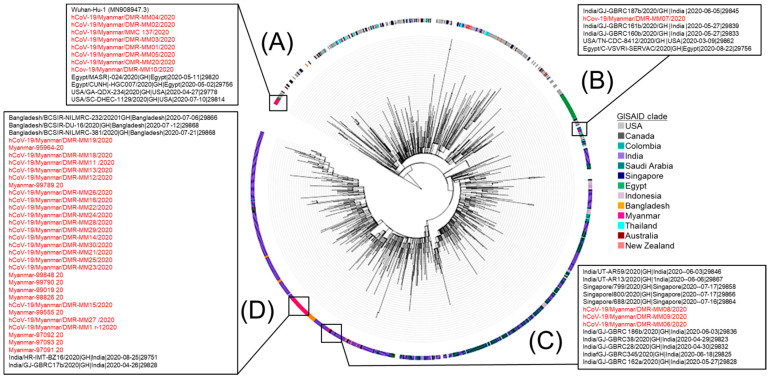
Phylogenetic tree of SARS-CoV-2 in Myanmar. Maximum-likelihood (ML) tree was created with 41 Myanmar strains, 509 global strains, and 770 strains in neighboring Asian countries collected from December 2019 to September 2020, downloaded from Global Initiative on Sharing All Influenza Data (GISIAD). Countries with more than 10 strains were used in the phylogenetic tree. The 41 strains from Myanmar are marked in pink, and different colors denote different countries. (**A**) Clade O, (**B**) Clade GR, (**C**) Clade GH, (**D**) Clade GH.

**Table 1 viruses-14-00259-t001:** Demographic characteristics of 10-COVID-19 patients in this study.

No	Date of RT-PCR Test	Township	Contact History	Travel History (Domestic)	Symptoms	Chronic Disease	Recovery
1	21 August 2020	Sittwe, Rakhine	Yes	No	No	No	Yes
2	23 August 2020	Sittwe, Rakhine	Yes	No	No	No	Yes
3	23 August 2020	Sittwe, Rakhine	Yes	No	No	No	Yes
4	23 August 2020	Sittwe, Rakhine	Yes	No	No	No	Yes
5	23 August 2020	Kyaukphyu, Rakhine	No	Yes	No	No	Yes
6	25 August 2020	Myebon, Rakhine	No	Yes (Sittwe on 22 August 2020)	Fever, loss of smell	No	Yes
7	25 August 2020	Maungdaw, Rakhine	No	Yes	No	No	Yes
8	25 August 2020	Maungdaw, Rakhine	No	Yes	No	No	Yes
9	25 August 2020	Kyauktaw, Rakhine	No	Yes	No	No	Yes
10	25 August 2020	South Okkalapa, Yangon	No	No	No	No	Yes

Notes: chronic diseases include hypertension, diabetes mellitus, chronic cardiac, and renal disease.

**Table 2 viruses-14-00259-t002:** List of nucleotide substitutions found in 10 strains from Myanmar in this study, as compared with reference strain Wuhan/WIV04/2019.

Nucleotide Position	241 5ʹUTR	3037 ORF1ab	11,083 ORF1ab, nsp6 (L37F)	14,408 ORF1ab	18,756 ORF1ab	18,877 ORF1ab	19,145 ORF1ab, nsp14 (S369F)	22,444 Spike	23,403 Spike (D614G)	25,563 ORF3a (Q57H)	26,735 M	28,854 N (S194L)
Name												
Wuhan/WIV04/2019	C	C	G	C	G	C	C	G	A	G	C	C
95964-20	T	T	T	T	T	T		T	G	T	T	T
97091-20	T	T	T	T	T	T	T	T	G	T	T	T
97092-20	T	T	T	T	T	T	T	T	G	T	T	T
97093-20	T	T	T	T	T	T	T	T	G	T	T	T
98826-20	T	T	T	T	T	T		T	G	T	T	T
99019-20	T	T	T	T	T	T		T	G	T	T	T
99555-20	T	T	T	T	T	T		T	G	T	T	T
99789-20	T	T	T	T	T	T		T	G	T	T	T
99790-20	T	T	T	T	T	T		T	G	T	T	T
99848-20	T	T	T	T	T	T		T	G	T	T	T

## Data Availability

The whole-genome sequences generated in this study have been deposited in the Global Initiative on Sharing All Influenza data (GISAID) EpiCoV coronavirus SARS-CoV-2 database and the accession numbers are listed in Appendix A.

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
