# Peer review of "Epidemiology and Genetic Analysis of SARS-CoV-2 in Myanmar during the Community Outbreaks in 2020"

_viruses, 2022, doi:10.3390/v14020259_

Round 1

Reviewer 1 Report

The manuscript is build around 10 new SARS-CoV-2 sequences from the second wave in Myanmar, obtained from the Rakhine State (n=9) and one from the Yangon Region. Some analyses are complemented with 31 sequences already available from GISAID and included in a previous publication (https://doi.org/10.1038/s41598-021-89361-7). The potentially most interesting analysis based on viral sequences (the phylogenetic tree in Supplementary figure S3) does not include those sequences, and the conclusions obtained for the second wave in Myanmar are based on a very low and most likely unrepresentative sample of 10 sequences. 
In my opinion, the study can be improved only in two ways, either by the addition of the clinical and epidemiological characteristics of the 31 sequences to the ML phylogenetic tree and the ensuing analyses or by removing the phylogenetic analysis and restricting the analyses to only the epidemiological component.

Reviewer 2 Report

The ongoing COVID-19 pandemic led to more than 300 billions of confirmed cases and almost 5,5 billion of deaths. In addition, the SARS-CoV-2 mutates, what causes the emergence of new genetic variants with potential different properties, i.e., transmission. Therefore, it is extremely important to monitor the genetic diversity of SARS-CoV-2. Phyo and colleagues in their manuscript presented the genetic diversity of novel coronavirus during the second pandemic wave in Myanmar and connect the data to official epidemiology data. The biggest limitation, described in the Discussion by authors, is relatively small number of analyzed sequence from patients. The manuscript is clear and well-written, and I do not have any comments and with great pleasure I recommend to accept the article in the current form.

Round 2

Reviewer 1 Report

The authors have addressed adequately my previous concerns. They have included additional sequences from Myanmar into their analysis and have completed a more thorough study. Nevertheless, they need to change the order and numbering of current figures 3 and 4 to accommodate their mention in the text.

This manuscript is a resubmission of an earlier submission. The following is a list of the peer review reports and author responses from that submission.